# Single-photon detection using large-scale high-temperature MgB$_2$ sensors at 20 K

Ilya Charaev [1,2] ✉, Emma K. Batson [1], Sergey Cherednichenko [3] ✉, Kate Reidy [1], Vladimir Drakinskiy[3], Yang Yu [4], Samuel Lara-Avila [3], Joachim D. Thomsen [1], Marco Colangelo [1,5], Francesca Incalza[1], Konstantin Ilin[6], Andreas Schilling[2] & Karl K. Berggren [1] ✉

Ultra-fast single-photon detectors with high current density and operating temperature can benefit space and ground applications, including quantum optical communication systems, lightweight cryogenics for space crafts, and medical use. Here we demonstrate magnesium diboride (MgB$_2$) thin-film superconducting microwires capable of single-photon detection at 1.55 $\mu$m optical wavelength. We used helium ions to alter the properties of MgB$_2$, resulting in microwire-based detectors exhibiting single-photon sensitivity across a broad temperature range of up to 20 K, and detection efficiency saturation for 1 $\mu$m wide microwires at 3.7 K. Linearity of detection rate vs incident power was preserved up to at least 100 Mcps. Despite the large active area of up to $400 \times 400\ \mu$m$^2$, the reset time was found to be as low as ~ 1 ns. Our research provides possibilities for breaking the operating temperature limit and maximum single-pixel count rate, expanding the detector area, and raises inquiries about the fundamental mechanisms of single-photon detection in high-critical-temperature superconductors.

Superconducting Nanowire Single-Photon Detectors (SNSPDs)[1] have become crucial for a variety of applications, including quantum optics[2-5] and security[2,6-8], deep-space communication[9], biomedical imaging[10-12], and light detection and ranging (LIDAR)[13-15]. SNSPDs have achieved record-breaking efficiencies: a broadband high-efficiency detection from the soft X-ray[16] to mid-infrared spectral range[17]; close to 100 % system detection efficiency[18-20]; sub-3 ps temporal resolution (timing jitter)[21]; and low noise (referred to as dark count rate) of $6 \times 10^{-6}$ cps[22]. Despite this significant progress, the detection system requires cooling to at least 4.2 K (and often lower) due to the low critical temperature, $T_C < 10$ K, of utilized superconductors (NbN, NbTiN, WSi, MoSi)[23]. These temperatures require costly and bulky cryocoolers that limit practical application. As a consequence, there has been a significant interest in

superconductors with higher critical temperatures that would allow for operation at elevated temperatures[24].

In addition to the critical temperature, another material-related limitation arises due to the kinetic inductance, $L_k$, of superconducting devices[25]. In order to achieve a high photon-detection efficiency, one typically uses superconducting films of thicknesses $\leq 10$ nm[26] and high-resistivity (disordered) superconductors[17]. As a result, the films exhibit large values of $L_k$ ($10^2$-$10^3$ pH/square) thereby limiting the detector reset time[25], particularly problematic for large-area SNSPDs. The high critical temperature ($T_C$) superconductors with low kinetic inductance may be regarded as viable alternatives for detectors.

In an attempt to increase the operational temperature, higher $T_C$ materials have been widely explored. Early studies of thin films have suffered from serious challenges[27] in the development of quantum

[1]Massachusetts Institute of Technology, Cambridge, MA 02139, USA. [2]University of Zurich, Zurich 8057, Switzerland. [3]Department of Microtechnology and Nanoscience, Chalmers University of Technology, Göteborg SE-41296, Sweden. [4]Raith America, Inc., 300 Jordan Road, Troy, NY 12180, USA. [5]Department of Electrical and Computer Engineering, Northeastern University, 360 Huntington Ave., Boston, MA 02115, USA. [6]Institute of Micro- and Nanoelectronic Systems, Karlsruhe Institute of Technology (KIT), 76187 Karlsruhe, Germany. ✉e-mail: ilya.charaev@physik.uzh.ch; serguei@chalmers.se; berggren@mit.edu

detectors based on high-$T_C$ superconductors. The lateral proximity effect[28] with internal constrictions and grain boundaries[29], non-uniform distribution of the superconducting order parameters across the structure[30], instability of superconducting parameters[31], sensitivity to baking, and chemical interaction during the fabrication all impose limitations on the capacity to replicate detectors in a consistent manner. Moreover, the reduction of sensor dimensions, especially crucial for the detection of infrared photons with low energy, may lead to a suppression of superconductivity in low- and high-$T_C$ materials[32]. This phenomenon could limit both the sensitivity and operating temperature of potential devices.

Very recently, attempts to realize SNSPDs using two-dimensional high-$T_C$ materials revealed single-photon response in superconducting nanowires made of thin flakes of $Bi_2Sr_2CaCu_2O_{8+\delta}$, with single-photon response up to 25 K[33,34]. One major drawback of exfoliated flakes, however, is the inability to scale up the detector's active area.

Magnesium diboride ($MgB_2$) is a promising candidate for single-photon detectors that has been pursued previously for SNSPDs with mixed success[35–37]. The low kinetic inductance ($L_k \sim 2$ pH/square[36]) and critical temperature of 39 K make $MgB_2$ attractive from practical viewpoints. In SNSPDs made of $MgB_2$ thin films with $T_C = 30$ K, single-photon response in the optical range has been demonstrated at $T \approx 10$ K[35,37] and at 5 K for the near-infrared[36,38]. However, the non-uniformity of thin films over large areas[39], relatively low critical temperature[40], and low switching current in $MgB_2$ thin films remain essential unsolved challenges. The latest progress in Hybrid Physical-Chemical Vapor Deposition (HPCVD) has illuminated a route to grow high-quality superconducting films of $MgB_2$ for single-photon detectors.

In this work, we demonstrate SNSPDs fabricated from high-quality $MgB_2$ films with an active area of hundreds of square micrometers, critical temperature approaching the bulk value, and single-photon sensitivity up to 20 K (Fig. 1). To achieve this result, we used a helium-ion-beam-based irradiation process[41] that additionally enabled saturated detection of 1.55 $\mu$m single photons at 3.7 K. In contrast to previous $MgB_2$ detector attempts, we chose the micro-scale SNSPD geometry (width of the wire, $W \approx 1$ $\mu$m) recently demonstrated in a number of reports using low-$T_C$ superconductors[42–45]. In comparison to nanowires ($W \approx 100$ nm), superconducting microwires have far larger switching currents and a lower total kinetic inductance, making them attractive for the fabrication of large-area detectors. We observed a reset time of ~1 ns enabling a count rate of 100 Mcps for a device with an active area up to $400 \times 400$ $\mu m^2$.

## Results

### Design and characterization of the $MgB_2$ detectors

To fabricate detectors out of $MgB_2$ films, we systematically optimized the Hybrid Physical-Chemical Vapor Deposition (HPCVD) process to produce relatively thin films with low surface roughness and high critical temperature. We concentrated on 5–10 nm thin films (Fig. 2b), where a high critical-current density can be achieved[36]. This thickness range also facilitates stronger absorption of incident light in the infrared range[46]. Thicknesses were verified using x-ray reflectivity (XRR) (See Supplementary Note 6) and scanning transmission electron microscopy (STEM). We also used cross-sectional energy dispersive x-ray spectroscopy (EDS) to identify and map elemental composition. Figure 2c shows the results of qualitative analyses of a superconducting $MgB_2$ film deposited on the 6H-SiC substrate (See Methods and Supplementary Note 1 for details). While the total thickness of the studied sample was $\approx 12$ nm, the superconducting core of $MgB_2$ was measured to be only $\approx 7$ nm, and observed to be sandwiched between two oxide layers, one on the top of the film (2.5 nm) and the other one at the substrate interface (1.5 nm). A thin layer (assumed to be boron) was observed between superconducting $MgB_2$ and the top oxide layer. Boron is difficult to detect due to its light atomic mass but has been successfully identified in this region in similar samples (See Supplementary Fig. 9). These samples exhibited surprising resilience to degradation during standard lithography, etching, and contact deposition processes as well as storage in nitrogen over several months. Other films (unpatterned) lasted over three years in a vacuum desiccator. However, similar films have shown degradation under similar storage and processing conditions, indicating that the problem of the reliability and robustness of such films remains unresolved.

We analyzed the sample surface using Atomic Force Microscopy (AFM), observing a surface micro-roughness (RMS) of less than 1 nm over several square micrometers of film area (Fig. 2a). This low roughness suggests a minimal variation of the superconducting energy gap across the structure which is crucial for avoiding constrictions and thus maximizing the switching current[47].

We used electron-beam lithography (EBL) to define a meander structure for SNSPDs with various dimensions, $1200 - 10200$ $\mu$m in length and $1 - 5$ $\mu$m in width, with a filling factor of 0.28 (Fig. 1a). For fabrication details and device geometries, see Methods and Supplementary Note 3. We used 12 magnesium diboride films with various thicknesses for our experiment. Out of these, six films underwent material analyses, while the remaining six were employed in the nanofabrication process to create detectors.

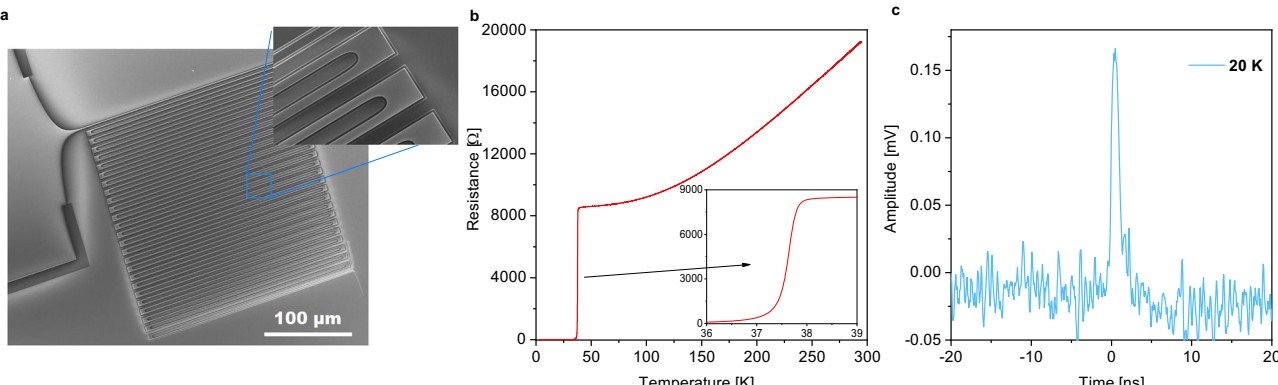

**Fig. 1 | Superconducting $MgB_2$ microwire-based single-photon detectors.**
**a** Scanning-electron microscopy (SEM) image of a 1 $\mu$m-wide meander-shaped microwire device. The smooth transitions of the microwires to the electrodes prevent current crowding. The scale bar is 100 $\mu$m. **b** Example of the $R(T)$ dependence of a 5 $\mu$m-wide meander-shaped microwire device. The measurement was done in a two-terminal configuration. The residual resistance ratio ($RRR$) was 2.25. **c** The voltage pulse from a photon event measured in the 1 $\mu$m $MgB_2$ microwire single-photon detector at 20 K and $\lambda = 1.55\mu$m. The device was biased to 98% of its switching current.

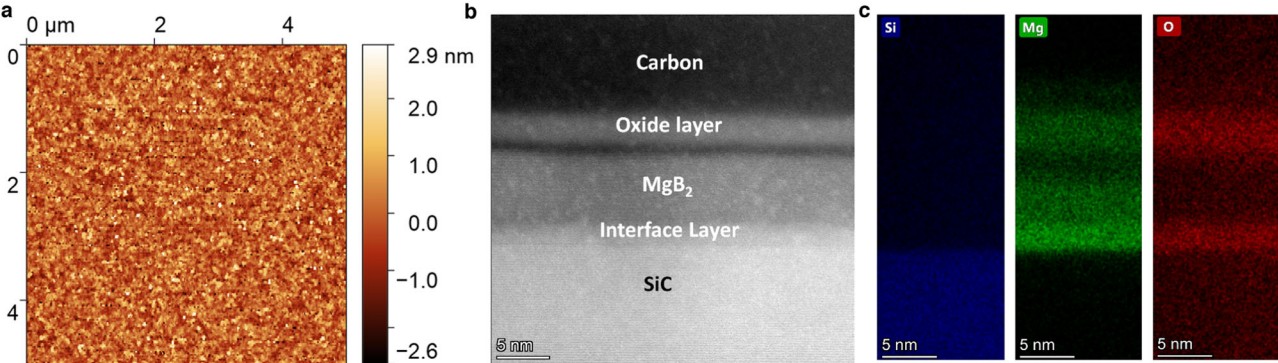

**Fig. 2 | Material analyses of superconducting MgB₂ films. a** Atomic-force microscope (AFM) image of the surface topography of a MgB₂ film (sample F180, Supplementary Fig. 1). The root-mean-square (RMS) surface roughness over several micrometers was less than 1 nm. **b** Cross-section of thin MgB₂ film taken via scanning transmission electron microscopy (STEM). Using focused-ion beam (FIB) preparation, a cross-section of the material was transferred onto a suitable TEM grid. The presence of carbon on top of the film is a result of the sample preparation.

**c** Energy-Dispersive X-ray Spectroscopy (EDS) for elemental characterization of Si, Mg, and O. The EDS spectrum provides the compositional information for the deposited MgB₂ films. Notably, the superconducting MgB₂ core is sandwiched between two oxides. While the MgB₂ films readily pick up oxygen from the ambient atmosphere forming oxides on the top, the presence of O₂ at the interface is likely from residual SiOₓ on the SiC substrate surface.

These devices, as initially fabricated, exhibited only dark counts. Motivated by our desire for single-photon detection we deliberately added defects by using 30-keV helium ion (He⁺) irradiation[41]. In contrast to patterning with heavier ions (e.g. gallium or xenon), the exposed regions were not observably etched by the He⁺ beam irradiation. The detector area was entirely exposed with He⁺ ions of $5 \times 10^{15}$ ions/cm² (See Supplementary Note 5). Such exposure had only a mild effect on switching current $I_{sw}$, $T_C$, and the normal state resistance $R_s$ (all changed by less than 5%).

After fabrication, we characterized the transport properties of our high-$T_C$ superconducting MgB₂ microwires. Figure 1b shows the temperature dependence of the resistance, $R(T)$, of a typical MgB₂ fabricated device, revealing the critical temperature of $37.6 \pm 0.3$ K as determined from the maximum of the $dR(T)/dT$. The obtained value is somewhat lower than those obtained for the parent MgB₂ film (38.7 K). This decrease indicates a mild degradation of the material's superconducting properties during the fabrication and ion beam irradiation. Additional information on superconducting and electrical properties can be found in Supplementary Note 4.

An important characteristic of single-photon detectors, enabling the generation of a voltage pulse upon single-photon absorption, is their metastable state that emerges under current biasing[48–50]. This metastable state is essential for detector operation and appears due to competition between the current-induced Joule self-heating of the microwire in the resistive state and electron cooling processes. This state manifests as a hysteresis in the $I$-$V$ characteristics. Figure 3a shows an example of an $I$-$V$ curve measured in a 5 μm-wide MgB₂ microwire device at a temperature (20 K) well below $T_C$ when the microwire is current-biased. A slight hysteretic behavior, characterized by the switching current $I_{sw} = 673$ μA and the retrapping current $I_r = 490$ μA, is observed.

We characterized the switching current of an array of devices by performing $I$-$V$ sweeps for wire widths from 1 μm to 5 μm at different temperatures; the switching current results are collected in Supplementary Note 4. They do not follow linear fits to the wire width, as one would expect the approaching zero, likely due to exceeding the width of the wire in relation to the Pearl length[51] (this value is ~1.3 μm for the penetration depth of 90 nm[36]) and non-uniform current distribution in the superconducting portion[51]. Notably, the deviation from the linear trend becomes more pronounced as the temperature decreases, aligning with the temperature-dependent penetration depth.

## Photoresponse measurements

To perform the photoresponse measurements, we mounted the chip with MgB₂ detectors in a variable-temperature cryostat equipped with RF coax cables and an optical fiber. The latter was held approximately 1 cm away from the device so that a defocused continuous wave laser beam covered the whole chip area. The simplified circuit diagrams that were used for the photoresponse measurements can be found in [see Fig. 3b[33]]. Detectors were measured in a conventional SNSPD configuration in which the device was biased through a DC input of the bias tee using an isolated voltage source with a bias resistor ranging from 1 to 10 kΩ. The AC output was connected to the low-noise amplifier whose output was fed to an oscilloscope or a photon counter. To mitigate latching effects, we used a low-bandwidth reset loop formed by a shunt inductor, $L_{sh}$, and resistor, $R_{sh}$[52] in parallel with the device.

Figure 1c shows an example of a generated photovoltage, measured across the current-biased MgB₂ microwires when the device was exposed to the laser beam radiation of wavelength $\lambda = 1.55$ μm. The traces of these devices shared common features with the photoresponse of conventional NbN SNSPDs. After reaching the maximum value within ~1 ns, the voltage exhibited a slower decay with the characteristic time $\tau$, often referred to as dead or recovery time, which depends on the total kinetic inductance, $L_k$, of the superconducting circuit and the load resistance[53]. The measured values (in particular, an example in Fig. 1c), $\tau \approx 1.3$ ns (determined as the time when the signal dropped to 30% of its maximum value) are in agreement with previous measurements of the kinetic inductance in MgB₂ films for different detector lengths at various temperatures[36]. The voltage spikes in MgB₂ devices were observed below and above liquid-helium temperature and could be detected up to $T = 20$ K. At higher temperatures, detectors did not exhibit a hysteresis, and thus no voltage pulses were observed upon illuminating the detectors with low-intensity laser light.

## Single-photon sensitivity of MgB₂ detectors

The MgB₂ devices exhibited single-photon sensitivity in the technologically-important 1550-nm telecommunications wavelength at the different powers of incident lights (Fig. 3b) and rates up to 100 Mcps. To obtain further insight into the performance of detectors, we recorded the photon count rate, $PCR$ (the number of pulses per unit time), as a function of the bias current, $I_b$. In these measurements, the light underwent an additional 40 dB of attenuation. Figure 3c shows the $PCR$ normalized to its maximum value, measured in MgB₂ devices

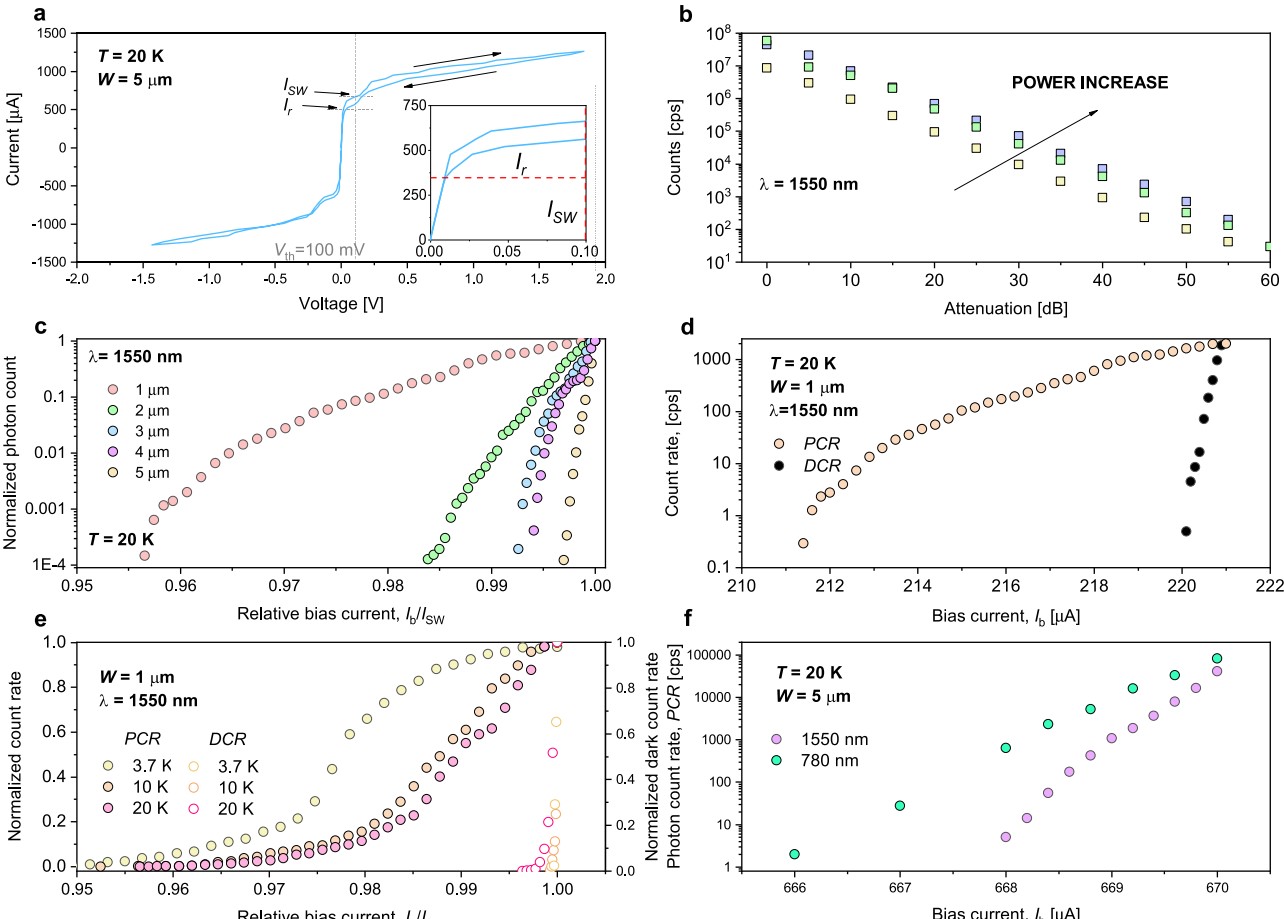

**Fig. 3 | Single-photon detection by superconducting MgB₂ microwires. a** *I*-*V* curve for 5 $\mu$m MgB₂ microwire detector measured at *T* = 20 K in the two-terminal configuration. **b** Photon count rate vs attenuation factor, at different powers of incident lights for 1-$\mu$m-wide MgB₂ device at given $I_b \approx 0.98I_{sw}$ and $\lambda = 1.55\,\mu$m. Measurement shows linearity over five orders of magnitude in count rates. **c** Normalized *PCR* vs relative bias current for different wire widths measured in the microscale MgB₂ devices at 20 K. **d** Photon and dark-count rate (*DCR*) of 1-$\mu$m wide

detector as a function of the absolute bias current ($I_b$) at 20 K. **e** The *PCR* and *DCR*, normalized to its maximum value, as a function of the relative bias current, $I_b/I_{sw}$, measured in 1-$\mu$m MgB₂ detector at given temperatures *T* and 1550-nm wavelength (λ). Notably, the count rate at 3.7 K is approaching a saturation plateau that suggests the internal detection efficiency of absorbed photons is approaching 100%. **f** *PCR* vs $I_b$ for different λ measured in 5-$\mu$m wide MgB₂ device at *T* = 20 K.

with widths of 1-5 $\mu$m upon exposing it to the $\lambda = 1.55\,\mu$m laser light at 20 K. Despite the biasing near the transition point at 99% of the switching current, the operational bias range remains in the micro-ampere range, making it similar to low-temperature detectors.

In the dark, spontaneous voltage pulses emerge when the device is biased close to the switching current (97-99%) (Fig. 3d), similar to low-*T* detectors. The absolute value of these dark counts did not exceed $2 \times 10^3$ s$^{-1}$, comparable to the values in conventional NbN SNSPDs. Upon illumination, the counts appeared at onset current $I_b = 0.96I_c$ whereas the *PCR* of the 1 $\mu$m wide device showed some tendency to saturation upon approaching $I_{sw}$ at 3.7 K (Fig. 3e). This saturation means high internal detector efficiency[54]. With increasing *T* to 20 K, the onset current decreased together with $I_{sw}$, as expected for superconducting devices (Fig. 3e). Furthermore, we found that the *PCR* for a given $I_b$ differed for photon energies corresponding to photons with $\lambda = 780$ nm and $\lambda = 1.55\,\mu$m (Fig. 3f). Single-photon operation was established through the observation of linear scaling of the photon count rate on the radiation power.

In the Supplementary Note 9, we provide estimates for the MgB₂ detector efficiency, *DE*. To this end, we used estimated absorption of thin films by determination of the impedance function[46]. We found that at 20 K, at which *PCR* is $\times 10^3$ s$^{-1}$ for the 40 dB attenuation, the *DE* is of the order of 7.6%. At 10 and 3.7 K, we find *DE* exceeding 10%, to be 11.3%, and 14.1% respectively. Note, that these estimates are

approximate as they do not account for the polarization dependence of meander geometry that can lead to both enhancement and reduction of the apparent efficiency. Nevertheless, we envision that *DE* can be boosted by integrating the MgB₂ detector into photonic cavities providing enhanced light-matter interaction.

We also found that the timing jitter of MgB₂ detectors is similar to that of conventional SNSPDs (~50 ps) (see Supplementary Note 8) with an active area at least one order of magnitude smaller than our devices. We used the 1 $\mu$m wide detector with an active area of 200 by 200 $\mu$m². This experiment was performed using the apparatus as described in the Supplementary Note 8.

## Discussion

While the experimental results show single-photon detection in microscale wire MgB₂ detectors at elevated temperatures several features of the data require discussion. The key aspects to discuss are (1) the details of the *I-V* characteristics after He⁺ exposure; (2) the mechanism of defect formation with He⁺ ion exposure; (3) the impact of the two-band character of MgB₂; and (4) the mechanism of photodetection in microwires; (5) the enhancement of the detection efficiency.

In Fig. 3, the IV-curve depicts multiple transitions occurring during the distinct jump from the superconducting to the resistive state. This observation on IV curves appeared after irradiation with He⁺ ions and was unexpected. This data supports the hypothesis of non-

uniform post-exposure with He$^+$ ions. The instability of the beam along with stitching error[55] is expected to result in variation of the superconducting gap over the detector area. We observed, at various temperatures and low bias currents, a noticeable increase in photon count rate with bias current which supports this hypothesis, suggesting localized areas with a lower gap and thus relatively higher internal detection efficiency. We furthermore expect that non-uniformity might have been introduced in the patterning and processing, therefore different regions of the wire could be participating in the detection process at different $T$ and $j_{bias}$. With process development including He$^+$ irradiation dose distribution correction, improved results may thus be expected.

The role of He$^+$ irradiation at various doses on MgB$_2$ films remains not fully explored. Drawing from the data reported for low-$T_C$ SNSPDs composed of NbTiN[56] and NbN[41], it's observed that even at He$^+$ dosages considerably lower than those necessary to induce noticeable effects on transport properties like critical current density or normal state conductivity, there is an enhancement in single-photon detection efficiency. In our study, we employed a dosage of $5 \times 10^{15}$ ions/cm$^2$ (equivalent to 50 ions/nm$^2$), a dose that aligns with the reported dosage required for 8-to-12 nm-thick NbTiN films to begin exhibiting single-photon response. This dose resulted in only modest variations, up to 10%, in parameters such as critical current density, critical temperature, and sheet resistance. These results closely approximate our findings. Nevertheless, comprehensive investigations in this area are still pending.

Apart from affecting MgB$_2$ film itself, the process of helium ion implantation causes defects and amorphization in the SiC substrate[57]. As a result, the efficiency of energy transfer from the superconducting structure to the substrate is likely to be affected. This effect may prolong the lifespan of the normal domain, thereby increasing the probability of detection before the normal domain collapses. In devices subjected to a relatively low dose of irradiation ($5 \times 10^{15}$ ions/cm$^2$), we noticed a decrease in the retrapping current. While the switching current remained unchanged, the retrapping current reduced to 90% of its nominal value compared to non-irradiated devices (Supplementary Fig. 7).

Although magnesium diboride has two bands with different gaps: $\pi$-band with a gap of $0.6 k_B T_C$ and $\sigma$-band with a gap of $2.2 k_B T_C$, the electron doping by atoms may initiate the interband scattering in MgB$_2$. The filling band effect increases the value of the small gap and, together with band filling, leads to the merging of the two gaps[58]. Our work points out that it is the uniformity of the superconducting gap, rather than the absolute gap, that is crucial.

The large bias current is not entirely unexpected in the case of high-$T_C$ microwire devices. The magnitude of the bias current is contingent upon the cross-sectional area of the wires and the critical temperature of the materials involved. In contrast to materials with lower critical temperatures typically employed in the production of superconducting nanowire single-photon detectors (SNSPDs), such as NbN, WSi, and MoSi, magnesium diboride has a critical temperature at least three times higher. That means that to achieve the same sensitivity, the gap must be suppressed by some means other than temperature, and elevating current density is one way to achieve a sufficiently small gap for detection.

It is also worth noting that, the width of wires surpassed the Pearl length in MgB$_2$. Wires wider than the Pearl length are expected to display non-uniform current distribution across the wire. The calculated Pearl length for MgB$_2$ films, approximately 1.3 $\mu$m, is four times lower than the widest wire used in our experiment. As a result, the ratio of the experimentally observed switching current density to the current density for spontaneous vortex entry is expected to degrade, proportionally to the following: $\sim 1/\sqrt{1 + W/2\pi \Lambda_P}$[59], where $\Lambda_P$ is the Pearl Length. Surprisingly, contrary to theoretical predictions[59], we observed single-photon detection in devices with wire widths of up to 5 $\mu$m. This observation challenges the previously known limit in wire

width from a practical standpoint, even though the mechanism of detection under these conditions remains unclear.

While the detection efficiency approaches the limit for a single-layer detector, primarily due to absorption in thin films, there exists an opportunity for further enhancement by integrating the detector into an optical waveguide. The integration of MgB$_2$ detectors onto a chip with a waveguide faces limitations imposed by the material that is used for the waveguide. The polycrystalline structure of magnesium diboride introduces lattice mismatch that impacts on superconducting and transport properties of films. On silicon carbide (SiC), it is feasible to produce MgB$_2$ films as thin as 3 nm, maintaining a critical temperature higher than 30 K with a narrow transition. Epitaxial growth, a crucial factor for achieving a high critical temperature in thin films, is employed in this process.

Although sapphire (Al$_2$O$_3$) has also proven to be a suitable substrate for MgB$_2$ films, the demonstration of MgB$_2$ films as thin as those on SiC remains unreported. Alternatively, the approach that involves light coupled through the substrate with an optical cavity positioned atop the detector can enhance the absorption efficiency in MgB$_2$ detectors.

## Methods

### Sputtering and fab

To fabricate detectors out of MgB$_2$ films we started with film deposition. Ultra-thin MgB$_2$ films were made using Hybrid Physical-Chemical Vapor Deposition (HPCVD) in a custom-built system[60,61]. Magnesium melts at 650 degrees Celsius and interacts with boron. The latter is produced via the thermal decomposition of gaseous diborane (B$_2$H$_6$) supplied in a mixture with hydrogen. Films were made on (0001) SiC substrates which provide excellent lattice matching to the hexagonal lattice of c-oriented MgB$_2$ films.

Devices were fabricated out of MgB$_2$ films on the 6H-SiC substrate with 5 nm titanium/ 50 nm gold contact pads using standard e-beam lithography, followed by metal deposition and lift-off in 45 °C N-Methylpyrrolidone (NMP). In order to produce the wires out of MgB$_2$ films, another round of e-beam lithography was performed, using 330-nm-thick high-resolution positive e-beam resist (ZEP520A). The patterns were then transferred onto the MgB$_2$ film by Ar$^+$ milling at a beam voltage of 300 V and Ar flow of 9 sccm for 12 minutes in a series of one-minute etches with three-minute-long intervals between each etch to avoid overheating of the sample. The residual resist was removed by immersing the substrate in the 45 °C NMP.

After fabrication and initial testing, we irradiated our samples with He$^+$ using a Zeiss Orion Microscope equipped with a Raith pattern generator. The irradiation was realized by sweeping the beam across the detector area. The exposed area was a rectangle with dimensions of active detector areas (to cover the whole meander area). The beam-limiting aperture was set to the largest diameter to maximize the beam current. The dose was varied in the range from $10^{15}$ to $10^{20}$ ions/cm$^2$ (Supplementary Fig. 5).

### Scanning Transmission Electron Microscopy (STEM)

STEM imaging was performed with a probe-corrected Thermo Fisher Scientific Themis Z G3 60–300 kV S/TEM operated at 200 kV with a beam current of 30-90 pA and 25 mrad convergence angle. Elemental analysis was accomplished by using EDS (Energy Dispersive X-ray Spectroscopy). For cross-sectional STEM characterization, samples were prepared using a Raith VELION focused ion beam-scanning electron microscope (FIB-SEM) system with carbon and platinum protection layers on top of the sample.

## Data availability

The data reported in Figs. 1–3 can be found on Zenodo: https://zenodo.org/records/10514254. The other data that support the findings of this study are available from the corresponding authors upon request.

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

## Acknowledgements
This work was carried out in part through the use of MIT.nano's facilities. I. Charaev acknowledges support for this work from Brookhaven Science Associates, LLC award number 030814-00001. E. Batson acknowledges support from the National Science Foundation under Grant No. EEC-1941583 and from the National Science Foundation Graduate Research Fellowship under Grant No. 2141064. This work was performed in part on the Raith VELION FIB-SEM in the MIT.nano Characterization Facilities (Award: DMR2117609). K.R. acknowledges funding and support from a MIT MathWorks Engineering Fellowship and ExxonMobil Research and Engineering Company through the MIT Energy Initiative. J.D.T. acknowledges support from the Independent Research Fund Denmark through Grant 9035-00006B. We also thank Boris Korzh (JPL) and Frances M. Ross for the helpful discussions. The work at Chalmers University of Technology was carried out with support from the Swedish Research Council (2019-04345) and the Swedish National Space Agency (198/16).

## Author contributions
I.C., S.C. and K.K.B. conceived and designed the project. I.C., E.B., S.L.A. and M.C. performed transport and photoresponse measurements. I.C. and E.B. fabricated devices. F.I. performed helium ion irradiation of some devices. I.C., S.C., E.B., K.I., A.S. and K.K.B. analyzed the experimental data. K.R. and Y.Y. performed STEM measurements. J.T. performed AFM measurements. S.C. deposited $MgB_2$ films. V.D. fabricated nanowires for transport measurements. I.C. wrote the manuscript with input from all co-authors. K.K.B. and S.C. supervised the project. All authors contributed to discussions.

## Competing interests
The authors declare no competing interests.
