## [Peer Review File · Nature Communications]

Single-photon detection using large-scale high-temperature MgB2 sensors at 20 KREVIEWER COMMENTS

Reviewer #1 (Remarks to the Author):

Dear authors.

The results presented in this article are original and they will be of interest for the community. The references are appropriate. The article is interesting and timely. However I have some points that, I think, should be addressed in order to clarify further the impact of this experiment, in particular for future applications of the detector.

1- Can you add an estimation of the detector quantum efficiencies at the different operating temperatures? Correctly you have claimed single photon detection capability up to 20k but only at low temperature (3.7K) a saturation behaviour (and hence a high DQE, needed for most applications) can be seen. How worse are the figure of merits at 20K?

2- It could be interesting to have in Fig 3e also the DCR for the three temperatures in order to visualize the operating ranges

3- For what concerns the timing jitter experiment please add the geometry of the detector (for example using table in fig S3) in both the main text and the text of the supplementary materials

4- Ref 32 and Ref 34 are not correctly reported

5- Superconducting MgB₂ film characterization in Supplementary materials: please report also the best parameters you have found for the deposition process (best flow, temperature, power ecc)

6- Second Critical magnetic field in Supplementary materials: please add the equation you have used to infer the value of D (or a citation).

7- He⁺ dose tests for MgB₂ detectors in Supplementary materials and fig. S6. It is not clear the role of those simulations. What have you inferred from those simulations? How those results compares with experimental data? Moreover the axes of fig s6 right are not clear and visible.

8- Timing jitter in Supplementary materials. Add in the text the device geometry referred to table reported in fig. s3

Reviewer #2 (Remarks to the Author):

The manuscript entitled "Single-photon detection using large-scale high-temperature MgB₂ sensors at 20K" by Charaev et. al. demonstrates the first single photon detection of large active area with a reset time as short as 1ns and an operating temperature of 20K. To achieve these results, the authors use high quality ultrathin MgB₂ film made by Hybrid Physical-Chemical Vapor Deposition and add defect to the microwires using Helium ion irradiation. The manuscript presents interesting new results regarding the development of superconducting single photon detectors. However, I have one major concern about the quality and validity of the evidence presented in the manuscript.

Currently, the main problem of MgB₂ for single photon detector is that the detection efficiency is very low, less than 0.01. The authors claim that the internal detection efficiency of the device is high, as the normalized count rate of 1 μ m wide device in Fig. 3e shows saturated behavior. However, the evidence of high detection efficiency is insufficient. The onset currents $I_b=0.96I_c$ in Fig. 3c and 3e are very large compared to devices using other materials. Further, the photon count rate appears to be very low from the intersection of PCR and DCR in Fig. 3d. More evidence is needed to verify the high detection efficiency. For example, the detection efficiency can be calculated from the illumination spot size or determined by focusing within a large device area and measuring again.

In summary, I think that the manuscript presents some novel results, but it needs to show the clear evidence of high detection efficiency to meet the standards of Nature Communications.

Reviewer #3 (Remarks to the Author):

In the manuscript titled "Single-photon detection using large-scale high-temperature MgB₂ sensors at 20 K" the authors a novel result wherein high-T_c, ultra-thin MgB₂ films deposited on 6H-SiC substrates are irradiated with 30 keV He-ions in order to induce defects in the film. This in turn enabled them to fabricate microns-wide SSPDs that showed near-IR single-photon sensitivity at temperatures as high as 20 K, while boasting recovery times and maximum count rates of 1.3 ns and 100 Mcps respectively in large-active area devices. Furthermore, the 1-micron-wide device showed PCR saturation for 1.55 μ m photons at a temperature of 3.7 K.

This work represents a major advance in the field of single-photon detection using superconductors and will be of interest to the wider community due to the new capabilities it can enable. Based off of the Journal's scope and existing published works, I am happy to recommend this manuscript for publication, provided the authors answer a handful of queries, and make some additions to the prose. I will present my reservations in three sections. The first one are questions that I would like addressed via additions or answered by way of reply. The second are optional changes to the manuscript. The third are minor editor-type errors that I picked up.

Section 1:

1. All decently impressive results with the MgB₂ material (like references 36-38) seem to deposit their films on lattice-matched SiC substrates. This appears necessary not just for quality superconducting films but also for thermal relaxation of the hotspot. Please comment on whether such recovery times and count rates are achievable on other, say, amorphous substrates. Elaborate in the discussion section on how this may limit applicability, as it disallows sandwiching between dielectric layers for efficient photon absorption, as well as use atop optical waveguides made of arbitrary material.

2. The IV-curve in fig. 3a is asymmetric. Is this significant. Also, how is I_{sw} defined if there are multiple jumps post He irradiation?

Section 2:

1. Page 1, left column, paragraph 1: References for SSPD applications in quantum optics must include some Bell's inequality violation results, as well as some certified-randomness generation and entanglement-swapping networks.

2. Page 2, left column, paragraph 2: More references are needed for microwires besides 43 & 44.

3. Page 3, left column, last paragraph: Along with reference 47, consider citing (a) J. K. W.

Yang et al., IEEE Trans. Appl. Supercond. 17, 581 (2007), and (b) A. Stockhausen et al., Supercond. Sci. Technol. 25, 035012 (2012).

4. Figure 3a: Add zoomed-in inset showing I_{sw} and I_r .

5. Page 4, left column, paragraph 1: Were the shunt elements off chip? And were they necessary for extracting ~ 1 ns pulses?

6. Page 5, right column, paragraph 1: It is mentioned that "parameters" showed "modest variations, up to 10%" when the devices were irradiated with He-ions, but there is only one figure presented contrasting non-irradiated with irradiated cases (fig. S7). Add more such to supplemental.

Section 3:

1. Re-arrange references 2-5 and 6-8 in chronological order.

2. Page 5, left column, last paragraph: insert a "that" between "expect" and "non-uniformity". And replace I_{bias} with J_{bias} , since you are referring to different segments of a single wire.

3. Page 6, left column, last paragraph: concatenate 'K' and 'V' in both occurrences.

4. References 25 and 27 are the same.

5. Reference 32 is U. R. Singh et al., Phys. Rev. B 88, 155124 (2013). You've only included the title.

6. Capitalize 'GeV' and 'MgB₂' appropriately in the titles of references 22 and 36 respectively.

7. Reference 53 has a stray 'U' inserted at the start of the title. And the journal details are missing (Advanced Quantum Technol., 2300139).

REVIEWER COMMENTS

Reviewer #1 (Remarks to the Author):

Dear authors.

The results presented in this article are original and they will be of interest for the community. The references are appropriate. The article is interesting and timely. However I have some points that, I think, should be addressed in order to clarify further the impact of this experiment, in particular for future applications of the detector.

We thank the Reviewer for their indication that our demonstration of single-photon detection using high-temperature superconducting MgB₂ microwires is original, timely, and interesting. Below we reply to all the Reviewer's comments/questions.

1- Can you add an estimation of the detector quantum efficiencies at the different operating temperatures? Correctly you have claimed single photon detection capability up to 20k but only at low temperature (3.7K) a saturation behaviour (and hence a high DQE, needed for most applications) can be seen. How worse are the figure of merits at 20K?

Given the dataset presented in this work, one can calculate the detector efficiency, DE , of our MgB₂ detectors by accounting for determined ~11.5% absorption (α) of thin MgB₂ films on a SiC substrate at 1550 nm wavelength by determination of the impedance function [Phys. Rev. B 80, 054510, 2009]:

$$DE = PCR [\alpha f A_d / A_{\text{beam}}]^{-1},$$

where A_{beam} is the area of the defocused laser beam incident on the chip, A_d is the total detector area, PCR is the photon count rate, and f is the photon flux. Using experimentally determined $PCR = 10^3 \text{ s}^{-1}$ for 1 μm wide detector (Fig. 3b) measured for the 40 dB attenuated laser radiation at $\lambda = 1500 \text{ nm}$ and assuming that the defocused laser beam spot radius is 75 μm , visually determined using red laser light fed into the same optical fiber, one obtains $DE \sim 7.6 \%$ at 20 K. At 10 and 3.7 K, we find DE exceeding 10 %, to be 11.3 %, and 14.1 % respectively. The discrepancy in values between DE and absorption arises marginally, primarily attributable to the reflection of photons from the holder back to the detector. However, these estimates are approximate as they do not account for the polarization dependence of meander geometry that can lead to both enhancement and reduction of the apparent efficiency. Moreover, we did not carry out accurate measurements of the A_{beam} and used the lower bound for the laser spot size. In addition, we did not account for variations in the power of the optical light source over a long time. All these adjustments may change the observed detection efficiency in either direction and, therefore, we originally refrained from reporting these numbers. In the revised Supplementary Material and Main Text, we included this information.

We note that maximizing the DE is beyond the scope of this study. This aspect could be explored in future publications, especially after a thorough investigation of the optical properties of thin MgB₂ films has been systematically conducted. By adding an appropriately designed dielectric

coating around the SNSPD, the DE could be raised to 98% (Optica 7, 1649 (2020)). Nevertheless, in comparison with the previous report on DE of MgB_2 SNSPDs, we have significantly enhanced the efficiency by two orders of magnitude.

2- It could be interesting to have in Fig 3e also the DCR for the three temperatures in order to visualize the operating ranges

We thank the Reviewer for this suggestion. In response, we have incorporated dark count rates for three different temperatures. Interestingly, in contrast to the temperature dependence observed in the photon count rate, the detectors exhibit a subtle variation in dark count rates with respect to current.

Fig. R1. Temperature-dependent photon and dark count rate of MgB_2 detector. Normalized photon and dark count rate in 1- μ m wide MgB_2 detector measured at different bath temperatures indicated in the legend.

3- For what concerns the timing jitter experiment please add the geometry of the detector (for example using table in fig S3) in both the main text and the text of the supplementary materials

We thank the Reviewer again for the suggestion. We used the 1- μ m wide detector with an active area of 200 by 200 μ m². Following the Reviewer's suggestion, the geometry of the detector in the jitter experiment was added to the main text and supplementary materials.

4- Ref 32 and Ref 34 are not correctly reported

These references were updated with the correct form.

5- Superconducting MgB₂ film characterization in Supplementary materials: please report also the best parameters you have found for the deposition process (best flow, temperature, power ecc)

Previous studies aimed to maximize the critical temperature in thin MgB₂ films and reported in reference [58] ([61] in the revised version). All parameters that were used in our experiment are available in the cited papers. Nevertheless, following the reviewer's suggestion, we added the following description to the Supplementary materials:

Magnesium melts at 650 degrees Celsius and interacts with boron. The latter is produced via the thermal decomposition of gaseous diborane (B₂H₆) supplied in a mixture with hydrogen (5%:95%) at a flow of 2 sccm with a background hydrogen flow at 400 sccm. The total pressure is 20 Torr.

6- Second Critical magnetic field in Supplementary materials: please add the equation you have used to infer the value of D (or a citation).

We used the following expression to estimate the diffusion constant:

$$D = - \frac{4k_B}{\pi e} \left(\frac{dB_{C2}}{dT} \right)^{-1}$$

The formula and references were added to the supplementary materials.

7- He+ dose tests for MgB₂ detectors in Supplementary materials and fig. S6. It is not clear the role of those simulations. What have you inferred from those simulations? How those results compares with experimental data? Moreover the axes of fig s6 right are not clear and visible.

In our study, damage per ion was estimated by using the SRIM [J. Ziegler, "SRIM and TRIM."] "quick calculations" mode to simulate vacancies in a target material stack resulting from irradiation with 30 keV helium ions. Although SRIM in this mode has some limitations, typically underestimating actual damage per ion [Current Opinion in Solid State and Materials Science, vol. 23, p. 100757, Aug. 2019.], this mode should be sufficient for qualitative comparisons of irradiation damage across different film stacks. The obtained values exhibit slight discrepancies when compared to the experimental results, attributable to fluctuations in both the material's parameters and the He ion beam.

We replaced Fig. S6 with another one with a larger size.

8- Timing jitter in Supplementary materials. Add in the text the device geometry referred to table reported in fig. s3

We thank the Reviewer for this suggestion. The corresponding information was added to the text.

Reviewer #2 (Remarks to the Author):

The manuscript entitled “Single-photon detection using large-scale high-temperature MgB₂ sensors at 20K” by Charaev et. al. demonstrates the first single photon detection of large active area with a reset time as short as 1ns and an operating temperature of 20K. To achieve these results, the authors use high quality ultrathin MgB₂ film made by Hybrid Physical-Chemical Vapor Deposition and add defect to the microwires using Helium ion irradiation. The manuscript presents interesting new results regarding the development of superconducting single photon detectors. However, I have one major concern about the quality and validity of the evidence presented in the manuscript.

We thank the Reviewer for acknowledging that our work represents new results in the development of single-photon detectors based on MgB₂. Below we reply to all the Reviewer’s comments/questions.

Currently, the main problem of MgB₂ for single photon detector is that the detection efficiency is very low, less than 0.01. The authors claim that the internal detection efficiency of the device is high, as the normalized count rate of 1µm wide device in Fig. 3e shows saturated behavior. However, the evidence of high detection efficiency is insufficient.

Given the dataset presented in this work, one can calculate the detector efficiency, DE , of our MgB₂ detectors by accounting for determined ~11.5% absorption (α) of thin MgB₂ films on a SiC substrate by determination of the impedance function [Phys. Rev. B 80, 054510, 2009]:

$$DE=PCR [\alpha f A_d / A_{\text{beam}}]^{-1},$$

where A_{beam} is the area of the defocused laser beam incident on the chip, A_d is the total detector area, PCR is the photon count rate, and f is the photon flux. Using experimentally determined $PCR=10^3 \text{ s}^{-1}$ 1 µm wide detector (Fig. 3b) measured for the 40 dB attenuated laser radiation at $\lambda=1.5 \text{ µm}$ and assuming that the defocused laser beam spot radius is 75 mm, visually determined using red laser light fed into the same optical fiber, one obtains $DE\sim 7.6 \%$ at 20 K. At 10 and 3.7 K, we find DE exceeding 10 %, to be 11.3 %, and 14.1 % respectively. The discrepancy in values between DE and absorption arises marginally, primarily attributable to the reflection of photons from the holder back to the detector. However, these estimates are crude as they do not account for the polarization dependence of meander geometry that can lead to both enhancement and reduction of the apparent efficiency. Moreover, we did not carry out accurate measurements of the A_{beam} and used the lower bound for the laser spot size. In addition, we did not account for variations in the power of the optical light source over a long time. All these adjustments may change the observed detection efficiency in either direction and, therefore, we originally refrained from reporting these numbers. In the revised Supplementary Material, we included this information.

We note that maximizing the DE is out of the scope of this work. By adding an appropriately designed dielectric coating around the SNSPD, the DE could be raised to 98% (Optica 7, 1649 (2020)). Nevertheless, in comparison with the previous report on DE of MgB₂ SNSPDs, we have

significantly enhanced the efficiency by two orders of magnitude. *Furthermore, we reached a limit of the DE for a single-layer detector due to the absorption in thin films.*

The onset currents $I_b=0.96I_c$ in Fig. 3c and 3e are very large compared to devices using other materials.

The reviewer is correct in pointing this out, and the reasons for this are not entirely understood. However, this large bias current is not entirely unexpected in the case of high- T_C microwire devices. The magnitude of the bias current is contingent upon the cross-sectional area of the wires and the critical temperature of the materials involved. In contrast to materials with lower critical temperatures typically employed in the production of superconducting nanowire single-photon detectors (SNSPDs), such as NbN, WSi, and MoSi, magnesium diboride has a critical temperature at least three times higher. That means that to achieve the same sensitivity, the gap must be suppressed by some means other than temperature, and elevating current density is one way to achieve a sufficiently small gap for detection. We have added a discussion of this issue to the main body of the paper.

In addition to the considerable energy gap, the current distribution along the width of the MgB₂ microwire is non-uniform, attributed to the small penetration depth. As a result, it necessitates biasing the microwire closer to the switching current when compared to superconducting wires with a uniform current distribution. Moreover, wide wires can be considered as parallel-connected nanowires. It is well-established that the threshold current in such configurations increases with the growing number of parallel-connected nanowires.

Further, the photon count rate appears to be very low from the intersection of PCR and DCR in Fig. 3d. More evidence is needed to verify the high detection efficiency. For example, the detection efficiency can be calculated from the illumination spot size or determined by focusing within a large device area and measuring again.

The photon count rate depicted in Fig. 3d is notably diminished, primarily attributed to the substantial attenuation of light and the relatively low power of incident light, measuring only a few micro-watts. Conversely, as evident in Fig. 3b, the count rate can surge to as high as 10^8 counts per second in the single-photon regime.

We thank the Reviewer for recommending the inclusion of detection efficiency. We have incorporated these calculations earlier in the manuscript as per the suggestion.

In summary, I think that the manuscript presents some novel results, but it needs to show the clear evidence of high detection efficiency to meet the standards of Nature Communications.

We believe we have shown evidence of high detection efficiency, and thus have satisfied this requirement.

Reviewer #3 (Remarks to the Author):

In the manuscript titled "Single-photon detection using large-scale high-temperature MgB₂ sensors at 20 K" the authors a novel result wherein high-T_c, ultra-thin MgB₂ films deposited on 6H-SiC substrates are irradiated with 30 keV He-ions in order to induce defects in the film. This in turn enabled them to fabricate microns-wide SSPDs that showed near-IR single-photon sensitivity at temperatures as high as 20 K, while boasting recovery times and maximum count rates of 1.3 ns and 100 Mcps respectively in large-active area devices. Furthermore, the 1-micron-wide device showed PCR saturation for 1.55 μm photons at a temperature of 3.7 K.

This work represents a major advance in the field of single-photon detection using superconductors and will be of interest to the wider community due to the new capabilities it can enable. Based off of the Journal's scope and existing published works, I am happy to recommend this manuscript for publication, provided the authors answer a handful of queries, and make some additions to the prose. I will present my reservations in three sections. The first one are questions that I would like addressed via additions or answered by way of reply. The second are optional changes to the manuscript. The third are minor editor-type errors that I picked up.

We thank the Reviewer for acknowledging that our demonstration of single-photon detection using high-temperature superconducting microwires is remarkable and supporting the publication of our manuscript. Below we reply to all the Reviewer's comments/questions.

Section 1:

1. All decently impressive results with the MgB₂ material (like references 36-38) seem to deposit their films on lattice-matched SiC substrates. This appears necessary not just for quality superconducting films but also for thermal relaxation of the hotspot. Please comment on whether such recovery times and count rates are achievable on other, say, amorphous substrates. Elaborate in the discussion section on how this may limit applicability, as it disallows sandwiching between dielectric layers for efficient photon absorption, as well as use atop optical waveguides made of arbitrary material.

The reviewer is correct in pointing out that ultra-thin films of MgB₂ so far have been reported on SiC. It is determined by the attempt to preserve a high T_c, which seems to be linked to epitaxial growth of MgB₂ on the lattice matched SiC. Since the electron energy relaxation time is inversely proportional to the electron temperature of the hot electrons ($T \approx T_c$) and directly proportional to the film thickness (PHYSICAL REVIEW B 94, 174509 (2016)), the criteria seem to be very reasonable. Furthermore, it has been shown, both experimentally and theoretically, that thinner superconducting films result in higher detection efficiency (J. Appl. Phys. 108, 014507 (2010)). Moreover, lattice defects in less structured (or amorphous) superconducting films lead to shorter electron mean free path, and hence to a higher L_k , which is another factor in utilizing SiC as the substrate in our present study.

The recovery time of SNSPDs is defined by the total kinetic inductance ($\sim R_s/T_c$) of the device and load resistance. Therefore, achieving a comparable count rate in MgB₂ detectors on amorphous substrates hinges on the interplay between the increase the critical temperature and the reduction of sheet resistance.

We also agree with the reviewer that limited choice of substrates for optical coupling schemes with MgB₂ photon detectors may restrict their overall applicability. However, this problem has not been properly studied yet. The published routes for fiber integrated SNSPDs are based on the material/substrate systems for the utilized films (WSi, NbN, MoSi, etc.). Certain routes may not be viable for MgB₂ photon detectors, while others remain applicable. For instance, one feasible approach involves light being coupled through the substrate, with an optical cavity positioned atop the detector. In our experience, growing Al₂O₃ on thin MgB₂ films without compromising their superconducting properties is achievable. Alternatively, dielectric films, apart from Al₂O₃, could be grown atop MgB₂. Another potential avenue involves exploring photonic waveguides out of SiC as an alternative integration method.

Following the reviewer's suggestion, we added the following section to the main text to discuss the integration of the MgB₂ detectors on the chip with a waveguide or dielectrical multilayer structure to enhance absorption:

While the detection efficiency approaches the limit for a single-layer detector, primarily due to absorption in thin films, there exists an opportunity for further enhancement by integrating the detector into an optical waveguide. The integration of MgB₂ detectors onto a chip with a waveguide faces limitations imposed by the material that is used for the waveguide. The polycrystalline structure of magnesium diboride introduces lattice mismatch that impacts on superconducting and transport properties of films. On silicon carbide (SiC), it is feasible to produce MgB₂ films as thin as 3 nm, maintaining a critical temperature higher than 30 K with a narrow transition. Epitaxial growth, a crucial factor for achieving a high critical temperature in thin films, is employed in this process. Although sapphire (Al₂O₃) has also proven to be a suitable substrate for MgB₂ films, the demonstration of MgB₂ films as thin as those on SiC remains unreported. Alternatively, the approach that involves light coupled through the substrate with an optical cavity positioned atop the detector can enhance the absorption efficiency in MgB₂ detectors.

2. The IV-curve in fig. 3a is asymmetric. Is this significant. Also, how is I_{sw} defined if there are multiple jumps post He irradiation?

The asymmetry in the IV curve suggests the presence of defects that can be induced by nonuniform irradiation with He⁺ or local current crowding near the edge of the wire. The temperature instability during the measurement time can probably cause asymmetry in the IV curves. While precise measurements are challenging, it is evident that, for identical values of the maximal positive and negative currents (1250 μA), the positive voltage substantially exceeds the negative counterpart. Therefore, positive Joule results in overheating of the samples and thus in a stronger hysteresis.

An important characteristic, enabling the generation of a voltage pulse upon single-photon absorption, is a metastable state that emerges under current biasing. This metastable state appears due to the competition between the current-induced Joule self-heating of the wires in the normal state and electron cooling processes and manifests itself as a pronounced hysteresis on the characteristics.

The critical current definition depends on the voltage threshold. This point corresponds to the 100 mV voltage threshold in our DC measurements (as shown in Fig. 3a) while in optical measurements we define the critical current without taking additional characteristics at the point where a pronounced voltage jump to the resistive state is observed. These values can be slightly different due to additional noise produced by readout circuits (amplifiers, bias-tee, and so on).

We thank the reviewer for the comments below as well and have made these changes. We also added comments to the questions below.

Section 2:

1. Page 1, left column, paragraph 1: References for SSPD applications in quantum optics must include some Bell's inequality violation results, as well as some certified-randomness generation and entanglement-swapping networks.

2. Page 2, left column, paragraph 2: More references are needed for microwires besides 43 & 44.

3. Page 3, left column, last paragraph: Along with reference 47, consider citing (a) J. K. W. Yang et al., IEEE Trans. Appl. Supercond. 17, 581 (2007), and (b) A. Stockhausen et al., Supercond. Sci. Technol. 25, 035012 (2012).

4. Figure 3a: Add zoomed-in inset showing I_{sw} and I_r .

5. Page 4, left column, paragraph 1: Were the shunt elements off chip? And were they necessary for extracting ~ 1 ns pulses?

The shunt inductor with a resistor was mounted on a cold stage (not on a chip). Due to low kinetic inductance 20-30 nH ($N_{sq}=10200$, as in Fig. S3), devices made of MgB_2 are often latched especially at the bias close to the switching current. To avoid unwanted latching effects at a high count rate and bias, a low-bandwidth reset loop was used in our experimental setup.

6. Page 5, right column, paragraph 1: It is mentioned that "parameters" showed "modest variations, up to 10%" when the devices were irradiated with He-ions, but there is only one figure presented contrasting non-irradiated with irradiated cases (fig. S7). Add more such to supplemental.

We thank the reviewer for the suggestion. We added examples of the variation of the critical temperature and resistance of the device (Fig. S7a).

Section 3:

1. Re-arrange references 2-5 and 6-8 in chronological order.

2. Page 5, left column, last paragraph: insert a "that" between "expect" and "non-uniformity". And replace I_{bias} with J_{bias} , since you are referring to different segments of a single wire.

3. Page 6, left column, last paragraph: concatenate 'K' and 'V' in both occurrences.
4. References 25 and 27 are the same.
5. Reference 32 is U. R. Singh et al., *Phys. Rev. B* 88, 155124 (2013). You've only included the title.
6. Capitalize 'GeV' and 'MgB_2' appropriately in the titles of references 22 and 36 respectively.
7. Reference 53 has a stray 'U' inserted at the start of the title. And the journal details are missing (*Advanced Quantum Technol.*, 2300139).

REVIEWERS' COMMENTS

Reviewer #1 (Remarks to the Author):

Dear authors.

The article is interesting and timely. Moreover this resubmitted version has been further improved helping the reader to compare this device to the others in literature. The points raised from the reviewers have been addressed.

Reviewer #2 (Remarks to the Author):

The authors calculate the detection efficiency of the device and show that it exceeds above 10%. They also include the discussion about the very high onset currents. This seems sufficient as an answer to my comments.

Reviewer #3 (Remarks to the Author):

The authors have satisfactorily addressed all of my concerns. The manuscript is ready and I recommend publication.

Minor grammatical error:

Page 5, left column, paragraph 2: The sentence, "This saturation indicates the approaching a high internal detector efficiency." is malformed.